# The Distinct Regulation of the Vitamin D and Aryl Hydrocarbon Receptors in COVID-19

**DOI:** 10.3390/nu16050598

**Published:** 2024-02-22

**Authors:** Oliver Robak, Marie-Theres Kastner, Astrid Voill-Glaninger, André Viveiros, Christoph Steininger

**Affiliations:** 1Department of Internal Medicine I, Medical University of Vienna, 1090 Vienna, Austria; marie-theres.kastner@meduniwien.ac.at (M.-T.K.); christoph.steininger@meduniwien.ac.at (C.S.); 2Department of Laboratory Medicine, Klinik Landstraße, 1030 Vienna, Austria; astrid.voill-glaninger@gesundheitsverbund.at (A.V.-G.); andre.viveiros@gesundheitsverbund.at (A.V.); 3Karl-Landsteiner Institute for Microbiome Research, Medical University of Vienna, 1090 Vienna, Austria

**Keywords:** VDR, COVID-19, ARDS, SARS-CoV-2, AhR, vitamin D

## Abstract

(1) Background: SARS-CoV-2 affects several immune pathways, including the vitamin D (VDR) and the aryl hydrocarbon receptor pathways (AhR). The aim of the study was the evaluation of the VDR and AhR pathways in the blood of COVID-19 patients with regard to the severity of disease. (2) Methods: Observational, single-center, case–control design. A total of 240 samples were selected for exploration. Patients who tested negative for SARS-CoV-2 but suffered from other respiratory infections (ORIs) served as a control group. (3) Results: VDR-specific mRNA in the blood of patients with mild symptoms (131.2 ± 198.6) was significantly upregulated relative to the VDR expression of the ORI group (23.24 ± 42.60; *p* < 0.0001); however, VDR expression of critically ill patients showed an impaired upregulation (54.73 ± 68.34; *p* < 0.001). CYP27B1 expression was not significantly regulated during SARS-CoV-2 infection. There was a downregulation of VDR and CYP27B1 compared to survivors. There was no significant difference in 25(OH)-vitamin D3 levels between critically ill patients with regard to survival (24.3 ± 9.4 vs. 27.1 ± 11.3; *p* = 0.433). (4) Conclusion: The VDR and AhR pathways are distinctively regulated in patients suffering from COVID-19 depending on the severity of disease. A combination treatment of antiviral drugs and vitamin D substitution should be evaluated for potentially improved prognosis in COVID-19.

## 1. Introduction

Severe acute respiratory syndrome coronavirus 2 (SARS-CoV-2) is a novel coronavirus that was discovered in January 2020 [1] after increasing numbers of patients suffering from severe respiratory failure were discovered in China at the end of 2019 [2]. SARS-CoV-2, a single-stranded RNA virus, is highly contagious in humans, and the cause of the ongoing coronavirus-induced disease (COVID-19) pandemic that has been declared an international public health emergency by the World Health Organization (WHO) [3,4].

As per the World Health Organization (WHO), the emergence of a new coronavirus variant known as SARS-CoV-2 towards the close of 2019 has triggered an unparalleled global pandemic, affecting an estimated 77 million individuals or more. Initial case reports emphasized the respiratory manifestations of the illness, including coughing, difficulty breathing, and, in severe instances, diminished oxygen levels necessitating hospitalization and mechanical ventilation [5]. Furthermore, SARS-CoV-2 infection causes pathologies in multiple organ systems which contribute to its unusually high lethality [6]. There is now consensus that a significant clinical challenge arises from a state of hyperinflammation triggered by SARS-CoV-2 infection in the early phases of the illness [7,8]. Inflammation is regulated by the interplay among various cell types, including granulocytes, macrophages, and endothelial cells [9,10,11], along with epithelial cells which then interact with distinct cells of the adaptive immune system [12]. Nevertheless, the current epidemiological data do not provide definitive answers to crucial questions regarding which microbiological, immunological, genetic, and metabolic factors play a significant role in shaping the progression of COVID-19.

The vitamin D receptor (VDR) is a crucial component in the regulation of various physiological processes, particularly those related to calcium homeostasis and bone health. However, emerging research suggests that VDR may also play a role in the context of infectious diseases, including COVID-19 [13,14]. The SARS-CoV-2 virus primarily targets the respiratory system, leading to a range of symptoms from mild respiratory distress to severe acute respiratory distress syndrome (ARDS). Vitamin D, synthesized in the skin or obtained through diet, is known to modulate the immune system [14,15]. VDR, when activated by vitamin D, influences the expression of genes involved in immune response and inflammation. Some studies propose that adequate levels of vitamin D could potentially enhance immune defense against viral infections, including COVID-19 [16,17]. Furthermore, VDR activation may contribute to the regulation of the hyperinflammatory response associated with severe COVID-19 cases [17]. While the relationship between vitamin D, VDR, and COVID-19 is an active area of investigation, it is essential to acknowledge the complex interplay of various factors influencing disease outcomes and the need for further clinical studies to establish a clear link between vitamin D status, VDR function, and susceptibility to or severity of COVID-19 [18]. While there is evidence that vitamin D status can impact the immune response and influence susceptibility to viral infections [18], the specific role of VDR gene expression differences between genders in viral infections is not well established. Research on the relationship between vitamin D and viral infections often focuses on the overall vitamin D status rather than gender-specific VDR gene expression. Some studies suggest that vitamin D deficiency may be associated with an increased risk of respiratory infections, including viral infections, and that supplementation may have a protective effect [18,19,20]. Regarding gender differences, it is known that there are sex-based variations in immune responses [21]. Females often exhibit a more robust immune response compared to males, which may influence the susceptibility and outcomes of viral infections [22,23,24]. However, the specific contribution of VDR gene expression differences between genders in this context is not thoroughly studied. Recently, we found that cytomegalovirus (CMV) infection downregulates VDR within hours after infection of mammalian cells in vitro and in vivo [25,26]. No such effect has been described for SARS-CoV-2 so far. 

The aryl hydrocarbon receptor (AhR) is a ligand-activated transcription factor that plays a crucial role in regulating various physiological processes, including xenobiotic metabolism, immune responses, and development. AhR is present in the cytoplasm in an inactive state. Upon binding to specific ligands, such as certain environmental pollutants, toxins, or naturally occurring compounds, AhR undergoes conformational changes, leading to its activation. Activated AhR translocates into the nucleus, where it forms a complex with other proteins. This complex acts as a transcription factor, binding to specific DNA sequences known as xenobiotic response elements (XREs) in the promoter regions of target genes. This binding regulates the expression of a wide range of genes involved in detoxification, metabolism, and immune responses.

We hypothesize that the VDR and AhR pathways are distinctively regulated in vivo during active SARS-CoV-2 infection with far-reaching clinical, immunological, and microbiological consequences.

## 2. Materials and Methods

### 2.1. Patients

The present study focused on VDR and its downstream effector pathways in COVID-19 patients in an observational, single-center, case–control design. The aim of the study was the evaluation of VDR, AhR, and dependent genes in these patients and correlation with virus load and disease severity. EDTA blood samples were routinely collected for this purpose at the time of first contact with the respective facility. Samples were cryopreserved at −80 °C. A total of 240 samples were selected for exploration. As the control group, we used patients who tested negative for SARS-CoV-2 but suffered from other respiratory infections (ORIs). 

Patients with mild COVID-19 symptoms and ORIs were recruited from the Emergency Department upon contact. Patients with severe symptoms were critically ill and were recruited from Intensive Care Units (ICUs). The criterium for a severe course of disease was whether the patients required oxygen due to respiratory impairment due to COVID-19 [27]. Sampling from the patients with severe symptoms took place upon admission at the ICU. Patients from the Emergency Department (with mild symptoms and ORIs) were sampled upon contact with the Emergency Department. Detailed demographic data (e.g., BMI) were only recorded in admitted patients. Patients from the control group (other respiratory infections—ORIs) and patients with mild COVID-19 symptoms (i.e., without the need for oxygen supplementation) were ambulatory, only seen briefly at the Emergency Department and soon discharged. Inclusion criteria comprised age > 18 and COVID-19-like symptoms (e.g., shortness of breath, fever, coughing); exclusion criteria comprised pregnancy and age < 18.

### 2.2. Ethics Statement

The current investigation adhered to the EU-GCP guidelines and complied with the principles outlined in the Declaration of Helsinki (1964), along with its subsequent revisions. Approval from the institutional review board was obtained (ECS# 1280/2020 and EK-20-099-VK, 31 March 2020).

### 2.3. Virology 

The cryopreserved samples were obtained during the initial hospital visit and were kept at −80 °C without thawing. PCR data for SARS-CoV-2 were accessible for all patients. Diagnosis of SARS-CoV-2 infection relied on the detection of SARS-CoV-2 DNA in nasal or oral swab samples confirmed by qPCR. The SARS-CoV-2 viral load was assessed using the Abbott RealTime SARS-CoV-2 qPCR assay on the m2000 RealTime platform, which includes automated DNA extraction by the m2000 sp instrument and real-time PCR by the m2000rt instrument (Abbott Laboratories in Abbott Park, IL, USA).

### 2.4. Isolation of RNA and Expression Screens

RNA was extracted from human patient EDTA blood samples utilizing the miRNeasy Mini Kit (QIAGEN, Hilden, Germany) and the RNase-Free DNase Set (QIAGEN, Hilden, Germany), following the manufacturer’s protocols. RNA concentration was assessed using a Nanodrop spectrometer (Nanodrop 8000 Thermo Scientific, Waltham, MA, USA), with an average yield of 100 ng/µL. An equal amount of eluted RNA was employed for all analyses. Total RNAs (15 ng/reaction) were reverse-transcribed using a mixture of random hexanucleotide primers from the iScript cDNA Synthesis Kit (Bio-Rad Laboratories, Hercules, CA, USA).

Quantitative reverse-transcription PCR (qRT-PCR) for screening the expression of genes associated with SARS-CoV-2 followed previously established protocols by our research team [25]. Samples (15 ng/reaction) were run in duplicates using Power SYBR Green PCR Master Mix (Life Technologies, Applied Biosystems, Vienna, Austria) on a Step One Plus qRT-PCR machine (Life Technologies, Applied Biosystems, Vienna, Austria). All values were normalized to the housekeeping gene ActB, and ΔΔCT was calculated relative to this reference gene [28]. Duplicates were performed, and No-Template Controls (NTCs) were included. The assay was conducted under standard conditions on a StepOnePlus instrument (Life Technologies). Analysis, normalization, and relative quantitation were performed using Microsoft Excel Professional Plus 19 and the ExpressionSuite Software v1.0.3 (Life Technologies).

### 2.5. Statistical Analysis

Statistical analysis of continuous data was carried out using the unpaired *t*-test with Welch’s correction. F distribution through the Kolmogorov–Smirnov test was utilized to detect normality. A two-sided *p* < 0.05 was considered statistically significant. The data are presented as mean and standard deviation or median and range. All statistical analyses were performed using the software GraphPad Prism 6.0 (GraphPad Software, San Diego, CA, USA) and SPSS 16.0 (SPSS Inc., Chicago, IL, USA). 

## 3. Results

A total of 240 patients were included, 132 (55.0%) of whom were female. Patient characteristics are depicted in Table 1. All patients were admitted to the hospital for COVID-19-like symptoms (e.g., coughing, fever, shortness of breath). Seventy-one (29.6%) patients were SARS-CoV-2-negative and suffering from other conditions (other respiratory infections, ORIs). 

To evaluate the impact of SARS-CoV-2 infection on the vitamin D system, we assessed the relative mRNA expression of VDR and 1α-hydroxylase (CYP27B1). Quantification of VDR-specific mRNA in the blood of patients with mild symptoms (131.2 ± 198.6) revealed that VDR expression was significantly upregulated relative to the VDR expression of the ORI group (23.24 ± 42.60; *p* < 0.0001); the upregulation was more pronounced in female patients but did not reach statistical significance (152.3 ± 71.36 vs. 110.8 ± 58.44; *p* = 0.162). VDR expression of critically ill patients showed an impaired upregulation (54.73 ± 68.34; *p* < 0.001; Figure 1A). Relative gene expression of CYP27B1 (the enzyme activating provitamin D) was quantified to evaluate positive or negative feedback loops associated with VDR expression. In contrast to VDR expression, CYP27B1 expression was not significantly regulated during SARS-CoV-2 infection (150.8 ± 535.3 and 132.1 ± 417.1) compared to ORI patients (77.17 ± 182.9; Figure 1C). When looking at the critically ill cohort who died, we found that there was a profound downregulation of both VDR and CYP27B1 compared to survivors (Figure 1B,D). There was no significant difference in 25(OH)-vitamin D3 levels on admission between critically ill patients who died and those who survived (24.3 ± 9.4 vs. 27.1 ± 11.3; *p* = 0.433, data incomplete). 

To assess the IFNγ pathway, we examined IFNγ, IL-12b, IL-1b, and STAT-1 and -3. Quantification of IFNγ-specific mRNA in the blood of patients with mild symptoms (47.95 ± 96.89) showed that IFNγ expression was not significantly different compared to the IFNγ expression of the ORI group (11.64 ± 15.90; *p* = 0.181); IFNγ expression of critically ill patients showed an upregulation (104.2 ± 347.8; *p* = 0.014; Figure 2A). STAT-1, which is downstream of IFNγ, was concomitantly upregulated in severe COVID-19 cases (11.28 ± 36.82 vs. 83.20 ± 288.3, *p*= 0.035), whereas STAT-3 did not show significant differences. Surprisingly, IL-1β was significantly downregulated in critically ill COVID-19 patients (36.27 ± 34.14) when compared to patients with mild symptoms (170.3 ± 362.5; *p* = 0.006, Figure 2B). There was no significant difference with regard to gender. 

AhR was significantly upregulated in patients with mild symptoms (56.59 ± 145.5; *p* = 0.015) and critically ill COVID-19 patients (37.61 ± 107.8; *p* = 0.016) when compared to ORI patients (8.118 ± 19.87; Figure 3A). We did not see significant differences in IL-22 and IL-17 in our exploration of these pathways. We saw a significant upregulation of IL-12 only in patients with mild symptoms compared to ORI patients (12.48 ± 36.32; *p* = 0.002; Figure 3B), whereas critically ill patients were lacking such upregulation (31.72 ± 93.78).

## 4. Discussion

Dysregulation of vitamin D metabolism and deficiency in vitamin D levels are common in critically ill patients and may impact outcomes significantly. We found in vivo evidence that VDR and its downstream pathways are differentially regulated during SARS-CoV-2 infection. This observation has far-reaching immunological and clinical implications and is in part contrary to previous data that showed a downregulation of VDR by SARS-CoV-2 and other viral pathogens [13,25]. 

Our principal finding indicates a rapid and significant increase in VDR mRNA levels in the blood of individuals experiencing mild symptoms of COVID-19. VDR is widely distributed throughout the human body, present in cells of various organs such as the pancreas, kidneys, intestines, bones, cartilage, epithelium, endocrine glands, and testes, as well as in T cells, monocytes, and dendritic cells [29]. Given VDR’s involvement in multiple immune signaling pathways, its expression not only affects vitamin D signaling but also influences the balance of other signaling pathways. Surprisingly, mRNA expression of 1α-hydroxylase (CYP27B1) was not dysregulated in these patients contrary to previous observations [25], indicating that either there are other VDR-dependent pathways involved or that the SARS-CoV-2 effect on VDR signaling is negligible. The significance of vitamin D levels during periods of SARS-CoV-2 infection and VDR regulation has to be clarified in this context as we did not have data on vitamin D levels in all of our patients. However, vitamin D substitution alone does not inhibit VDR downregulation in vitro during SARS-CoV-2 infection and also does not improve prognosis in solid-organ transplantation or HSCT [25]. Also, the lack of upregulation in critically ill patients might be due to two reasons. First, as shown in Table 1, the time from symptom onset and admission to the ICU and therefore sampling was much longer than in the group with ORI and mild symptoms. We might simply observe the natural course of disease here—upregulation in the early phase with a consecutive return to normal levels over time. Second, lack of VDR upregulation in critically ill patients might happen due to immune exhaustion in this group. 

The topic of distinct immune regulation with regard to gender is unfortunately a neglected one. It is known that there are sex-based variations in immune responses. Females often exhibit a more robust immune response compared to males, which may influence the susceptibility and outcomes of viral infections. When investigating a possible effect of gender on VDR regulation, we found a more pronounced upregulation in female patients with mild symptoms, which is in line with previous data [14,30]; however, this effect did not reach statistical significance. IFNγ was not distinctively regulated [31]. 

AhR signaling is a common host response to infection by multiple coronaviruses [32]. While the primary focus of AhR research has been on its role in response to environmental toxins and regulation of xenobiotic metabolism, recent studies have suggested potential connections between the AhR pathway and viral infections, including COVID-19 [33], and was recently identified as a host factor for Zika and dengue viruses [34]. It has been reported that although some NF-κB signaling is needed for coronavirus replication, excessive activation of this pathway may be deleterious for the virus [35]. AhR limits NF-κB activation, and interferes with multiple anti-viral immune mechanisms, including IFN-I production [36,37]. We found a mild upregulation of AhR in COVID-19 patients, which is in line with previous data [32]. AhR also activates immunological responses and inhibits inflammation through upregulation of IL-22 and downregulation of Th17 response [38], which we did not observe in our data. IL-12 mediates enhancement of the cytotoxic activity of NK cells and CD8+ cytotoxic T lymphocytes and was only upregulated in patients with mild symptoms, probably indicating an effective early response. 

IFNγ is predominantly produced by natural killer (NK) cells but also by other specialized cells of the immune system. Its production is controlled by positive regulators (e.g., IL-12) secreted mostly by antigen-presenting cells. IFNγ signals through the IFNγ receptor (IFNGR) and downstream activation of the JAK/STAT pathway [39]. IFNγ expression was not significantly different compared to the ORI group, which is not surprising since these patients were suffering from other respiratory infections, which, in most cases, were viral infections. Critically ill patients showed an upregulation of both IFNγ and STAT-1, as a surrogate parameter of an ongoing severe inflammatory response. 

Uncontrolled IL-1β production can be an underlying factor for ARDS in SARS-CoV-2 infection in the lung. Blood neutrophils are augmented by IL-1β and induce the overproduction of neutrophil extracellular traps (NETs), which can cause thrombus formation, epithelial and endothelial cell destruction, worsening of sepsis and ARDS, and ultimately multiple organ failure [40,41]. Contrary to other data [42], IL-1β production was impaired in critically ill patients, indicating that a robust IL-1β response might be pivotal for containment of COVID-19. Again, the difference might be explained by the characteristics of our cohort of critically ill patients showing immune exhaustion due to long-lasting inflammation.

A limitation of the present study is that we could not measure SARS-CoV-2 replication, VDR regulation, and innate immunity on a single-cell level, since there is evidence for a distinct role of VDR in CD4^+^ T cells [43]. The whole-blood samples contained multiple different cells; not all are permissive to SARS-CoV-2 infection and not all permissive cells usually harbor replicating viruses. Accordingly, the relative changes measured for VDR expression represent the means of the total. The patients were young from a medical point of view, with a mean of 48 (18–79) years of age; therefore, older, more vulnerable patients were underrepresented. Most patients (excluding the critically ill cohort) were ambulatory, meaning SARS-CoV-2 was detected rather late, since these patients are not as frequently tested as inpatients. We found no significant difference in 25(OH)-vitamin D3 levels on admission between critically ill patients who died and those who survived—albeit at a low level in both groups (not shown, data incomplete), which is in accordance with previous data [44]. VDR downregulation during COVID-19 has been described before [13]. However, VDR was significantly lower in critically ill non-survivors, which has not been shown to our knowledge so far. It would further be interesting to match these findings with published VDR gene polymorphisms (e.g., Fok I, Bsm I, Apa I, and Taq I) [45,46]. To check if VDR expression returns to normal levels, a second timepoint after recovery would have been ideal; however, this was not feasible in this setting.

## 5. Conclusions

In summary, in vivo VDR regulation in COVID-19 patients may further contribute to the well-known epidemiological link between this viral infection and following opportunistic infections. In the light of these findings, it is important to point out that a SARS-CoV-2 infection is a modifiable risk factor because of the availability of prevention strategies like vaccinations or face masks. We therefore propose the evaluation of a combination treatment with antiviral drugs together with vitamin D substitution for potentially improved prognosis in COVID-19. The insights gained from this study may open completely new avenues to an optimized treatment of such patients.

## Figures and Tables

**Figure 1 nutrients-16-00598-f001:**
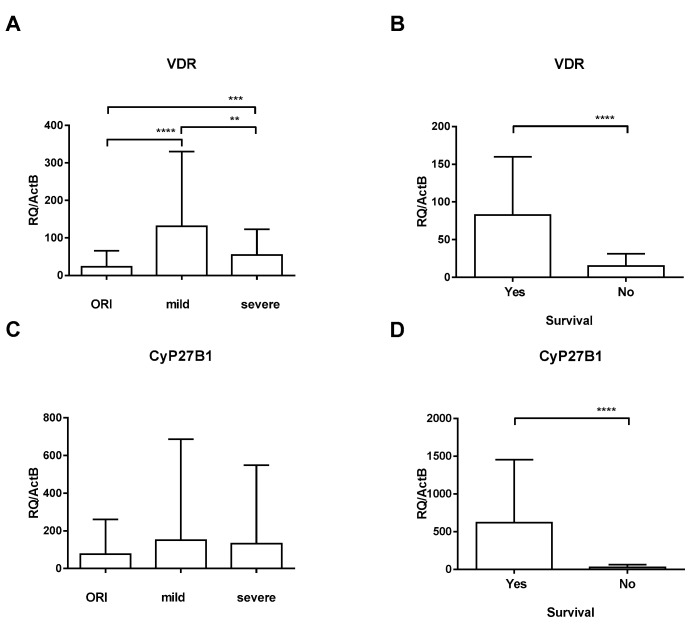
Expression of VDR (**A**,**B**) and CYP27B1 (**C**,**D**) was analyzed by RT-qPCR relative to ActB in patients with other respiratory infections (ORIs), patients with mild symptoms (mild), and critically ill ICU patients (severe). Error bars show mean and standard deviation. Asterisks show levels of significance (*p* > 0.05: Not significant (no asterisk), 0.01 < *p* ≤ 0.05: Significant (**), 0.001 < *p* ≤ 0.01: Highly significant (***), *p* ≤ 0.001: Very highly significant (****)).

**Figure 2 nutrients-16-00598-f002:**
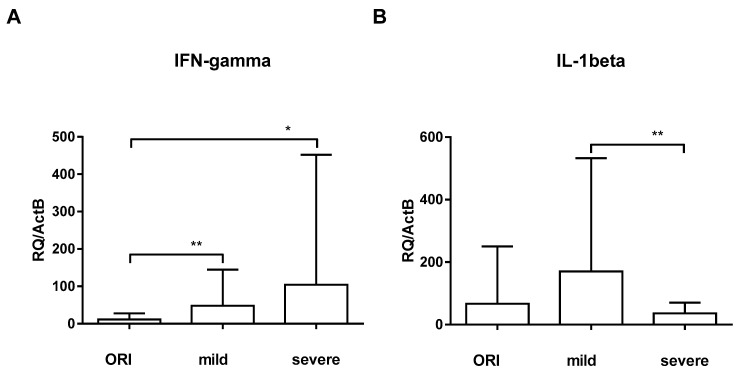
Expression of IFNγ (**A**) and IL-1β (**B**) was analyzed by RT-qPCR relative to ActB in patients with other respiratory infections (ORIs), patients with mild symptoms (mild), and critically ill ICU patients (severe). Error bars show mean and standard deviation. Asterisks show levels of significance (*p* > 0.05: Not significant (no asterisk), 0.01 < *p* ≤ 0.05: Significant (*), 0.001 < *p* ≤ 0.01: Highly significant (**), *p* ≤ 0.001: Very highly significant (***)).

**Figure 3 nutrients-16-00598-f003:**
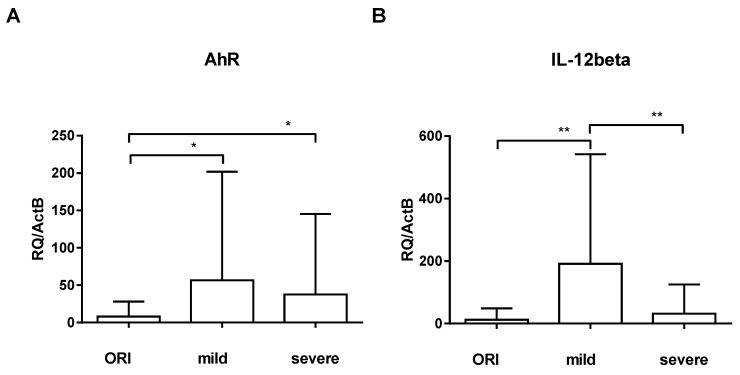
Expression of AhR (**A**) and IL-12β (**B**) was analyzed by RT-qPCR relative to ActB in patients with other respiratory infections (ORIs), patients with mild symptoms (mild), and critically ill ICU patients (severe). Error bars show mean and standard deviation. Asterisks show levels of significance (*p* > 0.05: Not significant (no asterisk), 0.01 < *p* ≤ 0.05: Significant (*), 0.001 < *p* ≤ 0.01: Highly significant (**), *p* ≤ 0.001: Very highly significant (***)).

**Table 1 nutrients-16-00598-t001:** Patients’ characteristics and demographics. BMI: body mass index, ICU: Intensive Care Unit, IMV: invasive mechanical ventilation, ECMO: extracorporeal membrane oxygenation, SAPS II: Simplified Acute Physiology Score II, SOFA: sequential organ failure assessment score.

			SARS-CoV-2-Infected			
	ORIs		Mild		Severe	
	N	%	N	%	N	%
Number of patients	71	29.6	64	26.7	105	43.8
Gender						
Male	31	43.7	30	46.9	47	40.0
Female	40	56.3	34	53.1	58	60.0
Median age in years	47		42		56	
(range)	(22–76)		(18–79)		(29–74)	
BMI on admission					23.1	
(range)					(20.0–33.1)	
Co-morbidities					N	%
Obesity (BMI ≥ 30 kg/m^2^)					53	50.5
Arterial hypertension					103	98.1
Diabetes mellitus					38	36.2
Ischemic cardiopathy					14	13.3
Cerebro-vascular disease					17	16.2
Chronic kidney failure					10	9.5
Chronic respiratory disease					45	42.9
Immunocompromised status					18	17.1
Main delays					Median	Range
Days between disease onset and hospital admission					2	(0–12)
Days between disease onset and ICU admission					8	(6–11)
Days between disease onset and intubation					10	(3–22)
Days between disease onset and ECMO (n = 77)					18	(4–36)
					Median	Range
Duration of IMV					15	(12–44)
Days between disease onset and IMV					6	(2–16)
Length of ICU stay, days					19	(12–45)
					N	%
On ECMO					77	37.4
Gender						
Female					27	21.3
Male					100	78.7
ICU Mortality					36	30.0
					Mean	SD
SAPS II					27.8	±3.4
SOFA					4.6	±1.9

## Data Availability

The data presented in this study are available on request from the corresponding author.

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
