# Peer review of "The Distinct Regulation of the Vitamin D and Aryl Hydrocarbon Receptors in COVID-19"

_nutrients, 2024, doi:10.3390/nu16050598_

Round 1
Reviewer 1 Report
Comments and Suggestions for Authors
The investigators detected the expression of VDR, AhR, and IFNγ, IL-12b, IL-1b, STAT-1 and -3 in the blood of COVID-19 patients. They report that VDR and its downstream pathways are differentially regulated. In addition, they found a mild upregulation of AhR in the course of SARS- CoV-2 infection.
Overall, this study presents the data in a logical manner, but there is a significant concern that would need to be addressed. Comments: - A gender difference in susceptibility, severity and mortality to COVID-19 has been observed (Ref. Russo C, Morello G, Malaguarnera R, Piro S, Furno DL, Malaguarnera L. Candidate genes of SARS-CoV-2 gender susceptibility. Sci Rep. 2021;11(1):21968. doi: 10.1038/s41598-021-01131-7). In paragraph 3 line 125, the authors report that 55% of patients were female, it would be interesting for the authors to highlight the differences in expression of the genes covered by this study between the two genera and to comment on their findings on this important aspect in the discussion. This would add more information about gender differences on the outcome of infection
- In the introduction line 64-67 References: Russo C, Valle MS, Malaguarnera L, Romano IR, Malaguarnera L. Comparison of Vitamin D and Resveratrol Performances in COVID-19. Nutrients. 2023;15(11):2639. doi: 10.3390/nu15112639; Malaguarnera L. Vitamin D3 as Potential Treatment Adjuncts for COVID-19. Nutrients. 2020;12(11):3512. doi: 10.3390/nu12113512 should be added

Author Response
The investigators detected the expression of VDR, AhR, and IFNγ, IL-12b, IL-1b, STAT-1 and -3 in the blood of COVID-19 patients. They report that VDR and its downstream pathways are differentially regulated. In addition, they found a mild upregulation of AhR in the course of SARS- CoV-2 infection.
- Overall, this study presents the data in a logical manner, but there is a significant concern that would need to be addressed. Comments: - A gender difference in susceptibility, severity and mortality to COVID-19 has been observed (Ref. Russo C, Morello G, Malaguarnera R, Piro S, Furno DL, Malaguarnera L. Candidate genes of SARS-CoV-2 gender susceptibility. Sci Rep. 2021;11(1):21968. doi: 10.1038/s41598-021-01131-7). In paragraph 3 line 125, the authors report that 55% of patients were female, it would be interesting for the authors to highlight the differences in expression of the genes covered by this study between the two genera and to comment on their findings on this important aspect in the discussion. This would add more information about gender differences on the outcome of infection
The topic of distinct immune regulation with regards to gender is unfortunately a neglected one. While there is evidence that vitamin D status can impact the immune response and influence susceptibility to viral infections, the specific role of VDR gene expression differences between genders in viral infections is not well-established. Research on the relationship between vitamin D and viral infections often focuses on the overall vita-min D status rather than gender-specific VDR gene expression. Some studies suggest that vitamin D deficiency may be associated with an increased risk of respiratory infections, including viral infections, and that supplementation may have a protective effect. Regarding gender differences, it's known that there are sex-based variations in immune responses. Females often exhibit a more robust immune response compared to males, which may influence the susceptibility and outcomes of viral infections. However, the specific contribution of VDR gene expression differences between genders in this context is not thoroughly studied. We added the data on gender differences: When investigating a possible effect of gender on VDR regulation, we indeed found a more pronounced upregulation in female patients with mild symptoms, which is in line with previous data; however, this effect did not reach statistical significance. We also did not find a significant difference in IFNγ. We thank the reviewer for enhancing the significance of our manuscript with this valuable insight.
- In the introduction line 64-67 References: Russo C, Valle MS, Malaguarnera L, Romano IR, Malaguarnera L. Comparison of Vitamin D and Resveratrol Performances in COVID-19. Nutrients. 2023;15(11):2639. doi: 10.3390/nu15112639; Malaguarnera L. Vitamin D3 as Potential Treatment Adjuncts for COVID-19. Nutrients. 2020;12(11):3512. doi: 10.3390/nu12113512 should be added
We thank the reviewer for these two interesting papers. We added them to the manuscript.
Reviewer 2 Report
Comments and Suggestions for Authors
The article “Distinct regulation of the vitamin-D and the aryl-hydrocarbon receptor in COVID-19” was aimed to evaluate VDR, AhR, and dependent genes in COVID-19 patients and correlation with virus load and disease severity. The methodology of the paper is poorly described and statistical analysis is unclear. I suggest to reject submitted article. Below present some of my remarks:
1. In methods section is poorly described, for example:
- I did not find crutial information about the measurement of basic antropometry and demographic data and explanation why some variables such as BMI was recorded only in group with severe disease (please see Table 1).
- I did not find crutial information about type of the unit/ward – from which patients were recruited. In discussion was written “as shown in Table 1, the time from symptom onset and admission to the ICU and therefore sampling was much longer than in the group with ORI and mild symptoms.” – I’am not sure whether each patient was recruited from Intensive Care Units, it wasn’t explicitly stated in method section.
- The part of Table 1 “Main delays” is unclear. In column “N” I found e.g. in row “Days between disease onset and ECMO (n=77)” that N = 18 and % = (4-36). What is the difference between n=77 and N=18 and what do you mean by the range given in column “%”
- Please interpret the numbers 4-36% and 18 from previous point in the result section. Most results of table 1 were not mentioned in the text.
- What was the definition “mild” and “severe” disease? Which criteria were used to devide patients to these two groups?
- I couldn’t find information about method of recruitment and any other details about included patients. The control group consisted of patients who tested negative for SARS-CoV-2 but suffered from other respiratory infections suggest that not only patient with COVID-19 disease were included.
- Design of the study is uclear. Here is control group but I found only that “in COVID19 patients in an observational, single-center, cohort design.”
- …….
2. Statistical part is unclear.
- The Authors wrote that “F-test was utilized to detect normality.” F-test is not dedicated to test normality. Could you present justification for the fact that analysed data had normal distribution? Generally such data are often very skewed.
- The Authors wrote that “unpaired t-test with Welch’s correction followed by Bonferroni’s post-test correction for multiple testing, if applicable.” “The data is presented as mean and standard deviation.”
- The Authors wrote that “unpaired t-test with Welch’s correction followed by Bonferroni’s post-test correction for multiple testing, if applicable.” T-test and Welch tests are applied to compare two groups – so how can you apply Bonferroni correction after them? To compare three groups Figure 1A, 1C, 2A, 2B – which test did you use?
- The Authors wrote that “The data is presented as mean and standard deviation.” Which data? For example Median age in years (range) is presented in table 1 – not mean and SD.
- What exactly do mean asterisks: “*”, “**”, “***” –
3. I did not find the explanation of many abbreviations such as ICU, invasive mechanical ventilation (IMV) or extracorporeal membrane oxygenation (ECMO).
4. The aim of the study in the abstract is not precise “The aim of the study was the evaluation of VDR, AhR, and dependent genes, in the blood of COVID-19 patients”, please compare it with the conclusion “Conclusion: The VDR and AhR pathways are distinctively regulated in patients suffering from COVID-19 depending on the severity of disease.”
Author Response
Comments and Suggestions for Authors
The article “Distinct regulation of the vitamin-D and the aryl-hydrocarbon receptor in COVID-19” was aimed to evaluate VDR, AhR, and dependent genes in COVID-19 patients and correlation with virus load and disease severity. The methodology of the paper is poorly described and statistical analysis is unclear. I suggest to reject submitted article. Below present some of my remarks:
- In methods section is poorly described, for example:
- I did not find crutial information about the measurement of basic antropometry and demographic data and explanation why some variables such as BMI was recorded only in group with severe disease (please see Table 1).
As we added to the methods section of the manuscript, BMI was only recorded in admitted patients. Patients from the control group (other respiratory infections – ORI) and patients with mild COVID-19 symptoms (i.e. without the need for oxygen supplementation, we also now clarified this in the methods section) were ambulatory, only seen briefly at the Emergency Department and soon discharged. Unfortunately, these data were not recorded by the physician in charge in these patients.
- I did not find crutial information about type of the unit/ward – from which patients were recruited. In discussion was written “as shown in Table 1, the time from symptom onset and admission to the ICU and therefore sampling was much longer than in the group with ORI and mild symptoms.” – I’am not sure whether each patient was recruited from Intensive Care Units, it wasn’t explicitly stated in method section.
Patients with mild COVID-19 symptoms and ORI were recruited from the Emergency Department upon contact and were discharged soon after. Patients with severe symptoms were critically-ill and were recruited from the Intensive Care Units. Sampling from the patients with severe symptoms took place upon admission at the ICU, in most cases patients have been sick for roughly 1-2 weeks. Patients from the Emergency Department (with mild symptoms and ORI) were sampled upon contact with the Emergency Department, which happened usually a couple of days after symptom onset. We clarified this in the methods section.
- The part of Table 1 “Main delays” is unclear. In column “N” I found e.g. in row “Days between disease onset and ECMO (n=77)” that N = 18 and % = (4-36). What is the difference between n=77 and N=18 and what do you mean by the range given in column “%”
“Main delays” refers to the time between critical timepoints during the course of the disease, e.g. the time from disease onset to the time until the first contact with the hospital. We further agree that using the number of patients (“N”) in this context was confusing and therefore corrected this. We also revised the labelling in the table itself to clarify our intent. 77 is the number of patients who received extracorporeal membrane oxygenation (ECMO). ECMO support was initiated at a median of 18 days after onset of disease. We hope that this revised table now enhances the readers’ understanding.
- Please interpret the numbers 4-36% and 18 from previous point in the result section. Most results of table 1 were not mentioned in the text.
With the former labeling, the numbers 4-36 and 18 were easily misinterpreted. As we clarified in the last paragraph, this does not refer to percentage but to median and range of days between disease onset and ECMO.
- What was the definition “mild” and “severe” disease? Which criteria were used to devide patients to these two groups?
The criterium for a severe course of disease was whether the patients required oxygen due to respiratory impairment due to COVID-19.
https://www.covid19treatmentguidelines.nih.gov/overview/clinical-spectrum/
We added the reference to the treatment guidelines to the manuscript.
- I couldn’t find information about method of recruitment and any other details about included patients. The control group consisted of patients who tested negative for SARS-CoV-2 but suffered from other respiratory infections suggest that not only patient with COVID-19 disease were included.
We now expanded on this in the methods section: Patients with mild COVID-19 symptoms and ORI were recruited from the Emergency Department upon contact. Patients with severe symptoms were critically-ill and were recruited from the Intensive Care Units (ICU).
- Design of the study is uclear. Here is control group but I found only that “in COVID19 patients in an observational, single-center, cohort design.”
We thank the reviewer for rightfully pointing this out. By design it was a case-control study. We are sorry for the confusion and corrected this in the manuscript.
- Statistical part is unclear.
- The Authors wrote that “F-test was utilized to detect normality.” F-test is not dedicated to test normality. Could you present justification for the fact that analysed data had normal distribution? Generally such data are often very skewed.
We thank the reviewer for his detailed review of the statistics section. To clarify, we used the F distribution test by the Kolmogorov-Smirnov test to detect normality. We changed this in the statistics section.
- The Authors wrote that “unpaired t-test with Welch’s correction followed by Bonferroni’s post-test correction for multiple testing, if applicable.” T-test and Welch tests are applied to compare two groups – so how can you apply Bonferroni correction after them? To compare three groups Figure 1A, 1C, 2A, 2B – which test did you use?
We rephrased the statistics section to clarify. In the mentioned figures, unpaired t-test with Welch’s correction was used due to the fact that the data were normally distributed but with unequal variances. Bonferroni’s post-test correction was not used because we only compared two groups as the reviewer rightfully points out. This has been deleted.
- The Authors wrote that “The data is presented as mean and standard deviation.” Which data? For example Median age in years (range) is presented in table 1 – not mean and SD.
We corrected this and clarified this in the respective table.
- What exactly do mean asterisks: “*”, “**”, “***”
We used asterisks to denote the level of statistical significance in the context of hypothesis testing and to represent p-values, which indicate the probability of obtaining the observed results or more extreme results if the null hypothesis is true:
p > 0.05: Not significant (no asterisk)
0.01 < p ≤ 0.05: Significant (*)
0.001 < p ≤ 0.01: Highly significant (**)
p ≤ 0.001: Very highly significant (***)
- I did not find the explanation of many abbreviations such as ICU, invasive mechanical ventilation (IMV) or extracorporeal membrane oxygenation (ECMO).
We checked the manuscript again for the respective abbreviations and clarified in the text.
- The aim of the study in the abstract is not precise “The aim of the study was the evaluation of VDR, AhR, and dependent genes, in the blood of COVID-19 patients”, please compare it with the conclusion “Conclusion: The VDR and AhR pathways are distinctively regulated in patients suffering from COVID-19 depending on the severity of disease.”
We rephrased the respective section in the abstract accordingly.
Reviewer 3 Report
Comments and Suggestions for Authors
The aim of the present study was the evaluation of VDR, AhR, and dependent genes in the blood of COVID-19 patients.
The topic of Covid is still a very interesting one.
I suggest the authors to take informations from this interesting article -
DOI 10.2147/RMHP.S284557
Please provide bibliography for this paragraph-Inflammation is governed by the 47 interaction of multiple different cell types such as granulocytes, macrophages, endothelial 48 along with epithelial cells which then interact with distinct cells of the adaptive immune 49 system. However, the current epidemiological data leave open the important questions of 50 which microbiological, immunological, genetic, and metabolic factors significantly influ- 51 ence the course of COVID-19.
Please specify what were the inclusion and exclusion criteria?
Please include more recent published articles in the discussion section.
Comments on the Quality of English LanguageModerate
Author Response
Comments and Suggestions for Authors
The aim of the present study was the evaluation of VDR, AhR, and dependent genes in the blood of COVID-19 patients.
The topic of Covid is still a very interesting one.
- I suggest the authors to take informations from this interesting article -
DOI 10.2147/RMHP.S284557
We thank the reviewer for this interesting paper.
- Please provide bibliography for this paragraph-Inflammation is governed by the 47 interaction of multiple different cell types such as granulocytes, macrophages, endothelial 48 along with epithelial cells which then interact with distinct cells of the adaptive immune 49 system. However, the current epidemiological data leave open the important questions of 50 which microbiological, immunological, genetic, and metabolic factors significantly influ- 51 ence the course of COVID-19.
We inserted citations for this paragraph to clarify. Inflammation is a complex process involving the interaction of various cell types, including granulocytes, macrophages, endothelial, and epithelial cells, as well as cells of the adaptive immune system (Cavender, 1991; Glaros, 2009; Kmieć, 2017). These interactions are mediated by a range of factors, such as cytokines and cellular receptors, and play a crucial role in immune response, tissue repair, and the pathogenesis of inflammatory diseases (Cavender, 1991; Glaros, 2009; Kmieć, 2017). Endothelial cells, in particular, are key players in this process, as they not only facilitate the migration of immune cells but also modulate their function (Danese, 2007). Understanding these interactions is essential for the development of targeted therapies for inflammatory conditions such as COVID-19.
- Please specify what were the inclusion and exclusion criteria?
Inclusion criteria comprised age >18 and COVID-19-like sympoms (e.g. shortness of breath, fever, coughing), exclusion criteria comprised pregnancy and age <18. We added this to the methods section of the manuscript.
- Please include more recent published articles in the discussion section.
We updated the bibliography to account for recent papers on this topic.
Round 2
Reviewer 2 Report
Comments and Suggestions for Authors
The Authors significantly improved manuscript and adressed all of my questions. I still have a minor remarks regarding e.g. Table 1. Please unify notation e.g. First we have Gender : Male Female and next on EMCO sex w and m. Please check and unify. Similarly not all columns in Table 1 have names (please add "N" at the top of the table befor "%" where needed
Author Response
- The Authors significantly improved manuscript and adressed all of my questions. I still have a minor remarks regarding e.g. Table 1. Please unify notation e.g. First we have Gender : Male Female and next on EMCO sex w and m. Please check and unify. Similarly not all columns in Table 1 have names (please add "N" at the top of the table befor "%" where needed
We have corrected table 1 accordingly, unified the notations, and re-arranged some data.